**Data Availability Statement:** All relevant data are within the manuscript and its Supporting Information files.

# Assessing depressive symptoms among people living with HIV in Yangon city, Myanmar: Does being a member of self-help group matter?

Myat Wint Than[1], Nicholus Tint Zaw[2], Kyi Minn[3], Yu Mon Saw[4,5], Junko Kiriya[1], Masamine Jimba[1], Hla Hla Win[6], Akira Shibanuma[1]*

1 Department of Community and Global Health, Graduate School of Medicine, University of Tokyo, Tokyo, Japan, 2 Innovation for Poverty Action Myanmar, Yangon, Myanmar, 3 Myanmar Health and Development Consortium, Yangon, Myanmar, 4 Department of Healthcare Administration, Nagoya University Graduate School of Medicine, Nagoya, Japan, 5 Nagoya University Asian Satellite Campuses Institute, Nagoya, Japan, 6 Department of Preventive and Social Medicine, University of Medicine 1, Yangon, Myanmar

* shibanuma@m.u-tokyo.ac.jp

## Abstract

### Background

While self-help groups have been formed among people living with HIV, few studies have been conducted to assess the role of self-help groups in mitigating depressive symptoms. This study investigated the association between self-help group membership and depressive symptoms among people living with HIV in Yangon, Myanmar.

### Methods

In this cross-sectional study, data were collected from people living with HIV at three antiretroviral therapy clinics in 2017. Multiple logistic regression analyses were carried out to examine the associations between having self-help group membership and depressive symptoms. Three ART clinics were purposively selected based on the recommendation from the National AIDS Program in Myanmar. At these clinics, people living with HIV were recruited by a convenience sampling method.

### Results

Among people living with HIV recruited in this study (n = 464), 201 (43.3%) were members of a self-help group. The membership was not associated with having depressive symptoms (adjusted odds ratio [AOR] 1.59, 95% confidence interval [CI] 0.98–2.59). Factors associated with having depressive symptoms were female (AOR 3.70, 95% CI 1.54–8.88) and lack of social support (AOR 0.97, 95% CI 0.96–0.98) among self-help group members, and female (AOR 3.47, 95% CI 1.70–7.09), lack of social support (AOR 0.98, 95% CI 0.97–0.99), and internalized stigma (AOR 1.28, 95% 1.08–1.53) among non-members.

**Funding:** This study was supported by the Asian Development Bank-Japan Scholarship Program. The funder had no role in study design, data collection and analysis, decision to publish, or preparation of the manuscript.

**Competing interests:** The authors report no conflicts of interest.

## Conclusions

This study did not find evidence on the association between membership in self-help groups and depressive symptoms among people living with HIV. Social support was a protective factor against depressive symptoms both self-help group members and non-members, although the level of social support was lower among members than non-members. The activities of self-help groups and care provided by the ART clinics should be strengthened to address mental health problems among people living with HIV in the study site.

## Introduction

As antiretroviral therapy (ART) access has been improved, HIV infection has become chronic, manageable condition [1]. Out of 37.9 million people living with HIV, nearly 23.3 million have access to ART globally in 2018 [2]. ART prolongs their survival [3]. However, psychological problems remain a concern for people living with HIV who disclose their status and seek care [4–6]. Depression is a common mental health disorder among people living with HIV [7,8]. People living with HIV have nearly twice the risk of having depressive symptoms compared to people without HIV [9,10]. This is a serious concern as people living with HIV with depression are more likely to engage in risky sexual behavior and to have suboptimal adherence to ART [11–13].

To overcome mental health and substance use problems, a self-help group approach has been encouraged [14]. In a self-help group, everyone has an equal opportunity to share problems and support each other [14,15]. They encourage each other to talk about their challenges, for example, stigma and discrimination, and advocate measures to address such challenges in the community [16]. By participating in a self-help group, people living with HIV can contact peers, form friendships, and exchange their knowledge and experiences [14,17,18]. Participation in self-help groups is likely to improve coping capacity through access to HIV-related knowledge and advice [19,20]. However, some people living with HIV avoid joining a self-help group, possibly because they want to hide their HIV-infection status [21].

Social support can play a vital role for people living with HIV in coping with depression [22]. It may work as a buffer against stress-related conditions. It may also boost psychological well-being among people living with HIV [23]. The higher the level of social support individuals receive, the lower the level of perceived stigma they face [24]. People living with HIV receive social support from family, friends, and co-workers [24,25], and it is sub-classified into actual support and perceived support [26]. Actual support addresses how family, friends, and co-workers intend to support people living with HIV. However, social support can be effective in coping with depression among people living with HIV once they perceive the support. Therefore, perceived social support may be a factor more directly related to depressive symptoms that people living with HIV have [26].

HIV infection is recognized as a public health issue in Myanmar [27]. In 2018, the estimated HIV prevalence was 0.8% among those who aged between 15 and 49 in Myanmar. Of the total 240,000 people living with HIV in Myanmar, 70.8% received ART in 2018 [28]. Despite these improvements in care, 60.0% of people living with HIV feel ashamed of being infected with HIV infection, and 18.0% are denied access to sexual and reproductive health services by health care providers due to their HIV status in Myanmar [29].

In Myanmar, as in other low- and middle-income countries, mental health interventions are not considered as a high priority compared to infectious disease control [30]. The

prevalence of depressive symptoms among people living with HIV was as high as 30.1% in a study conducted Yangon city, Myanmar [31]. Like other marginalized community, people living with HIV need to rely on their families and friends to cope with stress and depression [32]. Little is known, however, about how people living with HIV cope with depression in Myanmar [31]. Studies are needed to address the level of supports that help people living with HIV to cope with depression.

Self-help groups are active and increasing in number in Myanmar [33]. Members engage in different roles supporting their peers in ART clinics. They volunteer as clinic registration staff or as peer-to-peer counselors. No study has yet assessed how self-help groups dealt with depressive symptoms among people living with HIV, and previous studies focused mainly on factors associated with ART adherence rates and the retention and attrition rates in ART programs in Myanmar [34–37]. Only one study has reported that the prevalence of depressive symptoms was (30.1%) among people living with HIV in Yangon city, Myanmar [31]. The study, however, did not measure participation in self-help groups, and its effect on managing depression remains unclear in Myanmar.

This study aimed to examine the association between self-help group membership and depressive symptoms among people living with HIV in Yangon city, Myanmar. It also examined the differences in socio-demographic, clinical, social support, and mental health factors associated with depressive symptoms between self-help group members and non-members.

## Methods

### Study design

A comparative cross-sectional study was conducted in Yangon city, Myanmar, in 2017. The data were collected from August to September 2017. Yangon city was selected as the study site since the city was expected to have a large number of key populations, particularly vulnerable with HIV and those who received ART, although sub-national statistics related to HIV are scarce in Myanmar [38]. A total of 23,347 people living with HIV joined self-help groups in Myanmar, and 6,339 of them were members of self-help groups in Yangon city in 2015 [39]. As approximately 40% of total people living with HIV on ART were living in Yangon region in 2015, several international non-governmental organizations (NGOs) and ten government health teams implement HIV-related services in Yangon city [39].

### Study setting

This study was conducted in the venues of ART clinics. The ART clinics included in this study offered HIV testing and prevention awareness sessions. When people were diagnosed with HIV positive, the clinic provided ART and counseling, such as pre-test, post-test and ART adherence counseling. ART clinics provided meeting space for self-help group members to discuss their challenges. Self-help groups were independent of ART clinics that provided treatment for them.

### Participants, recruitment, and data collection

The required sample size was calculated using G*power (version 3.1) with a power of 80% and two-sided confidence level of 95%, based on the assumption following the practice of a previous study in the USA [40]. That study presented the mean and the standard deviation (SD) for depression as 9.5 and 8.3 among self-help group and 12.4 and 12.8 for standard care group. The calculation provided a required sample size of 444 in total (222 in each group). This study included people living with HIV aged 18 years or above who were enrolled in ART clinics at

Yangon city. Among them, this study excluded those who were co-infected with tuberculosis for the purpose of preventing further transmission of infections to interviewers. It also excluded critically ill people living with HIV who could not answer the questionnaire.

In this study, recruitment process was as follows: First, three ART clinics were purposively selected through consultation with the National AIDS Program (NAP). Assistant Director from NAP recommended three ART clinics that cooperated well with NAP and sent a regular report to NAP. Second, people living with HIV were recruited by a convenience sampling method when they visited the clinics during the data collection period. The clinic reception staff announced this study in the waiting area. Those who were interested in the participation received explanations from the staff on the purpose of this study. Only those who were willing to participate in this study were sent to the interview rooms. People living with HIV who participated in this study received soap and towel at the end of the interview. A structured questionnaire was administered through face-to-face interviews in Burmese language by trained interviewers in private rooms at the ART clinics.

## Exposure: Self-help group membership

This study measured 1) being a member of a self-help group and 2) the length of their membership in years [41]. This study identified self-help group members who had been involved in self-help groups for more than one year.

## Outcome: Presence of depressive symptoms

This study used the Hopkins Symptom Checklist for Depression (HSCL-D) to measure the severity of depressive symptoms [30,32]. The Burmese version of HSCL-D (Cronbach's alpha: 0.85) has already been validated among Myanmar refugees along the Thai-Myanmar border. HSCL-D showed satisfactory combined test-retest/inter-rater reliability (r = 0.84) and good internal consistency (0.92) [42].

An answer to each question was given on a four-point Likert-type scale (1 = not at all, 2 = a little, 3 = quite a bit, 4 = extremely). Item scores were added up and divided by the total number of answered items. The minimum mean score was 1, and the maximum mean score was 4. A mean score greater than 1.75 (cut-off point) was considered indicative of having depressive symptoms [30,32]. The same cut-off point (1.75) was applied to predict depressive symptoms among Myanmar refugees and people living with HIV in Cambodia [30,32,43]. In this study, the Cronbach's alpha was 0.85.

## Other covariates

Socio-demographic factors, HIV-related characteristics, perceived social support, and the type of a clinic were collected from people living with HIV as covariates in this study. The socio-demographic factors included age, gender, marital status, level of education, and main occupation [31]. HIV-related characteristics consisted of ART status and adherence, most recent CD4 count, and internalized HIV-related stigma. ART status was measured based on the interview questions on the regular intake or not, the frequency of intake, and the number of tablets per intake. ART adherence was measured based on an interview question on any missing of intake in the previous week of data collection. For the internalized HIV-related stigma, the AIDS Related Stigma Scale was used. Each item was rated dichotomously (1 = agree or 0 = disagree); with a total score ranging from 0 to 7. A Cronbach's alpha for this scale in the Burmese version was 0.80 [35]. In this study, a Cronbach's alpha was 0.68.

For measuring perceived social support, the Burmese version of the 19-item Medical Outcome Study Social Support Survey (MOS-SSS) was used; its Cronbach's alpha was 0.91 [44,45].

Responses to individual items ranged from one (none of the time) to five (all the time); total scores ranged from 0 to 100. The Cronbach's alpha coefficients for subscales in the Burmese version were 0.87 on the emotional subscale, 0.85 on the tangible subscale, 0.87 on the affectionate subscale, and 0.82 on the positive social interaction subscale [45]. In this study, Cronbach's alpha was 0.96 for this scale.

In addition, the variable of the type of clinic was recorded as the organization of the ART clinic where people living with HIV was recruited. It was used as dichotomized: NGO clinic and other (clinic run by NAP or a community-based organization).

## Statistical analyses

A Chi-square test or Fisher's exact test (in case of having at least one of the cells in a table had the expected value of 5 or below) was used for categorical variables, and the *t*-test was used for continuous variables to examine the association between each of these variables and self-help group membership status.

Multiple logistic regression analysis was conducted to find factors associated with having depressive symptoms. As the covariates of the regression model, only statistically significant variables at 5% level found in bivariate logistic regression analyses were included among variables for HIV-related characteristics, perceived social support, and the type of a clinic. In addition, all of the socio-demographic factor variables mentioned in the previous sub-section were included as covariates. Moreover, stratified with membership status, multiple logistic regression analyses were conducted to examine the differences in socio-demographic, clinical, social support, and mental health factors associated with depressive symptoms. No multicollinearity was detected. All statistical tests were two-sided with significance level at 5%. All statistical analyses were performed using STATA SE Version 13.1 (Stata Corporation, TX).

## Ethics considerations

This study was approved by both the Research Ethics Committee of the Graduate School of Medicine at the University of Tokyo (11647) and the Ethics Review Committee of the Department of Medical Research in Myanmar (Ethics/DMR/2017/111). Participation in this study was voluntary, and all PLHIV were given explanations of the study's objectives, procedures, benefits, confidentiality, and their right to withdraw from participation in this research. After PLHIV agreed to participate in this study, they were asked to sign an informed consent form prior to the interview. Confidentiality was maintained by not recording any personal identifiers in all instruments and analyses.

## Results

### Participants characteristics

A total of 469 people living with HIV were recruited and interviewed at three ART clinics. One of them did not finish the interview, and four people living with HIV were excluded because they knew they had tuberculosis. As a result, 464 people living with HIV were included in data analyses. Of these 464 people living with HIV, 201 had joined a self-help group (members), and 263 people living with HIV had not (non-members). The mean age was 39.2 (SD 9.5) years among members and 38.7 (SD 9.6) years among non-members. Of the members, 26.9% were men, 55.2% were women, and 17.9% were LGBT/unmentioned gender identity. Of non-members, 37.3% were men, 53.9% were women, and 8.8% were LGBT or did not identify their gender identity (Table 1).

Depressive symptoms were found among 89 members (44.3%) and 86 non-members (32.7%). The mean score of internalized stigma was 0.6 (SD 1.1) among members, which was

**Table 1. Socio-demographic characteristics of people living with HIV (N = 464).**

| Characteristics | Self-help group members (n = 201) | | Non-members (n = 263) | | p-value |
|---|---|---|---|---|---|
| | n (%) | Mean (SD) | n (%) | Mean (SD) | |
| **Age** | | 39.2 (9.5) | | 38.7 (9.6) | 0.526[b] |
| **Gender** | | | | | |
| Men | 54 (26.9) | | 98 (37.3) | | **0.003[a]** |
| Women | 111 (55.2) | | 142 (53.9) | | |
| LGBT/unmentioned gender identity | 36 (17.9) | | 23 (8.8) | | |
| **Marital status** | | | | | |
| Single/separate/widow | 132 (65.7) | | 121 (46.0) | | **<0.001[a]** |
| Married/living together | 69 (34.3) | | 142 (54.0) | | |
| **Education** | | | | | |
| Primary school/informal education | 44 (21.9) | | 80 (30.4) | | **0.007[a]** |
| Middle school | 52 (25.9) | | 84 (31.9) | | |
| High school and above | 105 (52.2) | | 99 (37.7) | | |
| **Main occupation** | | | | | |
| Dependent | 44 (21.9) | | 84 (31.9) | | **0.001[a]** |
| Employee | 90 (44.8) | | 101 (38.4) | | |
| Own business | 47 (23.4) | | 71 (27.0) | | |
| Other * | 20 (9.9) | | 7 (2.7) | | |
| **Type of clinic** | | | | | |
| NAP | 59 (29.4) | | 7 (2.6) | | **<0.001[c]** |
| NGO | 133 (66.1) | | 250 (95.1) | | |
| CBO | 7 (3.5) | | 4 (1.5) | | |
| Other** | 2 (1) | | 2 (0.8) | | |
| **Taking ART currently** | | | | | |
| No | 4 (2.0) | | 5 (2.0) | | 1.000[c] |
| Yes | 197 (98.0) | | 258 (98.0) | | |
| **Missed ART last month** | | | | | |
| No | 192 (97.5) | | 249 (96.5) | | 0.561[a] |
| Yes | 5 (2.5) | | 9 (3.5) | | |
| **Know CD4 count** | | | | | |
| No | 44 (21.9) | | 96 (36.5) | | **0.001[a]** |
| Yes | 157 (78.1) | | 167 (63.5) | | |
| **Comorbidity** | | | | | |
| No | 156 (77.6) | | 224 (85.2) | | **0.036[a]** |
| Yes | 45 (22.4) | | 39 (14.8) | | |
| **Exercise** | | | | | |
| No | 80 (39.8) | | 162 (61.6) | | **<0.001[a]** |
| Yes | 121 (60.2) | | 101 (38.4) | | |
| **CD4 Count** | | 612.8 (260.1) | | 553.8 (281.4) | 0.051[b] |
| **Social support** | | 53.4 (29.5) | | 65.1 (32.1) | **<0.001[b]** |
| **Internalized stigma** | | 0.6 (1.1) | | 1.0 (1.5) | **0.008[b]** |
| **Depressive symptoms** | | | | | |
| No | 112 (55.7) | | 177 (67.3) | | **0.011[a]** |
| Yes | 89 (44.3) | | 86 (32.7) | | |

SD: Standard deviation; LGBT: Lesbian, Gay, Bisexual and Transgender; NAP: National AIDS Program; NGO: Non-governmental organization; CBO: Community-based organization; ART: Antiretroviral therapy.

* Other included female sex workers, pensioners, faith-based volunteers, community-based volunteers and students.

**Other included General Practitioner, paid private clinics.

[a] Chi-square test

[b] t-test

[c] Fisher's exact test.

lower than the mean score of non-members, 1.0 (SD 1.5). The mean perceived social support score was 53.4 (SD 29.5) among members and 65.1 (SD 32.1) among non-members (Table 1).

## Factors associated with depressive symptoms

Being members was not associated with having depressive symptoms (adjusted odds ratio [AOR] 1.59, 95% confidence interval [CI] 0.98–2.59) (Table 2). Female members (AOR 3.61, 95% CI 2.14–6.11) and LGBT/unmentioned gender identity (AOR 2.36, 95% CI 1.14–4.90) were at higher risk of having depressive symptoms than male members.

## Factors associated with depressive symptoms among members

Female members were at higher risk of having depressive symptoms than male members (AOR 3.70, 95% CI 1.54–8.88). Those with middle school (AOR 0.19, 95% CI 0.07–0.57) and high school or higher education status (AOR 0.25, 95% CI 0.08–0.71) were at lower risk of

**Table 2. Factors associated with having depressive symptoms among people living with HIV (N = 464).**

| Variables | AOR | 95% CI | p-value |
|---|---|---|---|
| **SHG member** | | | |
| (Ref = No) | 1.00 | | |
| Yes | 1.59 | 0.98–2.59 | 0.061 |
| **Age** | 1.00 | 0.98–1.02 | 0.902 |
| **Gender** | | | |
| (Ref = men) | 1.00 | | |
| Women | **3.61** | **2.14–6.11** | **<0.001** |
| LGBT/unmentioned gender identity | **2.36** | **1.14–4.90** | **0.021** |
| **Marital status** | | | |
| (Ref = Single/Separate/Widow) | 1.00 | | |
| Married/Living together | 1.46 | 0.89–2.38 | 0.137 |
| **Education** | | | |
| (Ref = Primary school/informal education) | 1.00 | | |
| Middle school | 0.59 | 0.34–1.06 | 0.078 |
| High school and above | 0.70 | 0.40–1.23 | 0.218 |
| **Main occupation** | | | |
| (Ref = Dependent) | 1.00 | | |
| Employee | 1.33 | 0.78–2.28 | 0.298 |
| Own business | 1.14 | 0.62–2.11 | 0.657 |
| Other* | 0.91 | 0.30–2.78 | 0.837 |
| **Type of clinic** | | | |
| **NAP, CBO** | 1.00 | | |
| **NGO** | 0.71 | 0.39–1.28 | 0.258 |
| **Exercise** | | | |
| (Ref = No) | 1.00 | | |
| Yes | 0.69 | 0.44–1.09 | 0.116 |
| **Social Support** | **0.98** | **0.97–0.98** | **<0.001** |
| **Internalized stigma** | **1.28** | **1.11–1.49** | **0.001** |

AOR: Adjusted odds ratio; CI: Confidence interval; LGBT: Lesbian, Gay, Bisexual and Transgender; ART: Antiretroviral therapy; NAP: National AIDS Program; NGO: Non-governmental organization; CBO: Community-based organization; ART: Antiretroviral therapy

* Other included female sex workers, pensioners, faith-based volunteers, community-based volunteers and students.

having depressive symptoms than those with primary school/informal education. Members who registered at NGO clinics were at lower risk of having depressive symptoms (AOR 0.39, 95% CI 0.18–0.83). Members who perceived more social support were at lower risk of having depressive symptoms (AOR 0.97, 95% CI 0.96–0.98) (Table 3).

### Factors associated with depressive symptoms among non-members

Female non-members were at higher risk of having depressive symptoms than their male counterparts (AOR 3.47, 95% CI 1.70–7.09). LGBT/unmentioned gender identity were at higher risk of having depressive symptoms than male members (AOR 3.15, 95% CI 1.01–9.77). Those who perceived more social support were at lower risk of having depressive symptoms (AOR 0.98, 95% CI 0.97–0.99). Those who felt more internalized stigma was at higher risk of having depressive symptoms (AOR 1.28, 95% 1.08–1.53) (Table 3).

## Discussion

Being members of self-help groups was not associated with having depressive symptoms in Yangon, Myanmar. However, self-help group members registered at NGO clinics had a lower

**Table 3. Factors associated with having depressive symptoms stratified by the membership status of a self-help group (N = 464).**

| Variables | Self-help group members (n = 201) | | | Non-members (n = 263) | | |
|---|---|---|---|---|---|---|
| | AOR | 95% CI | p-value | AOR | 95% CI | p-value |
| **Year in SHG** | 1.02 | 0.92–1.15 | 0.617 | | | |
| **Age** | 0.98 | 0.95–1.02 | 0.279 | 1.01 | 0.98–1.05 | 0.280 |
| **Gender** | | | | | | |
| (Ref = Men) | 1.00 | | | 1.00 | | |
| Women | **3.70** | **1.54–8.88** | **0.003** | **3.47** | **1.70–7.09** | **0.001** |
| LGBT/unmentioned gender identity | 2.21 | 0.75–6.47 | 0.148 | **3.15** | **1.01–9.77** | **0.047** |
| **Marital status** | | | | | | |
| (Ref = Single/Separate/Widow) | 1.00 | | | 1.00 | | |
| Married/Living together | 1.13 | 0.52–2.46 | 0.752 | 1.44 | 0.71–2.91 | 0.301 |
| **Education** | | | | | | |
| (Ref = Primary school/informal education) | 1.00 | | | 1.00 | | |
| Middle school | **0.19** | **0.07–0.57** | **0.003** | 0.97 | 0.46–2.01 | 0.940 |
| High school and above | **0.25** | **0.08–0.71** | **0.009** | 1.09 | 0.52–2.28 | 0.803 |
| **Main Occupation** | | | | | | |
| (Ref = Dependent) | 1.00 | | | 1.00 | | |
| Employee | 1.45 | 0.56–3.77 | 0.437 | 1.37 | 0.69–2.75 | 0.363 |
| Own business | 2.05 | 0.71–5.85 | 0.180 | 0.77 | 0.34–1.76 | 0.548 |
| Other * | 0.38 | 0.72–2.03 | 0.260 | 2.98 | 0.51–17.14 | 0.220 |
| **Type of clinic** | | | | | | |
| NAP, CBO | 1.00 | | | 1.00 | | |
| NGO | **0.39** | **0.18–0.83** | **0.015** | 1.98 | 0.41–9.45 | 0.389 |
| **Exercise** | | | | | | |
| (Ref = No) | 1.00 | | | 1.00 | | |
| Yes | 0.72 | 0.54–1.60 | 0.381 | 0.62 | 0.33–1.18 | 0.148 |
| **Social Support** | **0.97** | **0.96–0.98** | **<0.001** | **0.98** | **0.97–0.99** | **0.003** |
| **Internalized stigma** | 1.27 | 0.94–1.72 | 0.111 | **1.28** | **1.08–1.53** | **0.004** |

AOR: Adjusted odds ratio; CI: Confidence interval; LGBT: Lesbian, Gay, Bisexual and Transgender; ART: Antiretroviral therapy; NAP: National AIDS Program; NGO: Non-governmental organization; CBO: Community-based organization; ART: Antiretroviral therapy.

*Other included female sex workers, pensioners, faith-based volunteer, community-based volunteers and students.

risk of having depressive symptoms compared to members at NAP/CBO clinics. In both groups, receiving more social support was a protective factor against depressive symptoms, although the mean level of social support was lower among self-help group members than non-members. Among non-members, feeling more internalized stigma was a risk factor for having depressive symptoms.

Being self-help group members was not associated with having depressive symptoms. However, members registered at NGO clinics had a lower risk of having depressive symptoms compared to members who registered at NAP/CBO clinics. The NAP/CBO clinics had fewer human and material resources than NGO funded clinics. Some members who registered at NAP/CBO clinics volunteered in these clinics, for example, for tracing the lost-to-follow-up of PLHIV [35,36]. This might cause stress and burden on the members [36]. Health professionals at NAP/CBO clinics may need an opportunity to learn from NGO clinics regarding how NGOs were supporting self-help groups.

The length of membership in the self-help group was not associated with having depressive symptoms. The membership in a self-help group is self-selective, not a discrete event such as attending a therapy session [46]. Some may participate in a group for many years, while others may drop out. The members decide individually how intensively, with what frequency, and what duration they want to participate in the group [46]. The quality of the membership experience may be more important than its length [47]. To improve the quality of self-help groups, members need to receive the opportunity for capacity building on enhancing decision making, promoting the social inclusion and legally registration of self-help groups [48].

In this study, gender differences existed in depressive symptoms among members and non-members, consistent with previous studies [49,50]. The prevalence of depression tends to be higher among women in both the general population and among people living with HIV. HIV-infected women and LGBT face stigma and discrimination in their daily lives [43,51]. Under persistent social norms related to gender, once women and LGBT are infected with HIV, they may perceive a higher level of stress than men. To address their stress, health professional needs to take proactive roles in raising gender equality awareness in the community and encouraging family members of people living with HIV to respect gender diversity [52].

Lower education level was associated with having depressive symptoms among members [31]. This association was not found among non-members. The effect of participating in self-help groups on depressive symptoms might be affected by members' education level. Members with middle school and high school or higher educations benefited more from self-help groups. Those who completed no more than primary school or informal education appeared to be disadvantaged in self-help groups. The self-help groups in this study may be dominated by people living with HIV with higher education levels [48].

Those who perceived more social support were at lower risk of depressive symptoms, both among members and non-members (Table 3). However, the mean score for perceived social support was significantly lower among members than non-members in this study (Table 1). Members supported each other by sharing their experience during self-help group meetings and invite people living with HIV to attend their meetings [53]. The people living with HIV who do not perceive social support tend to join self-help groups to seek more support from their peers [26].

Non-members who felt more internalized stigma were more likely to have depressive symptoms. However, this association was not found among the members. The prevalence of internalized stigma was lower among the members of self-help groups than non-members. Internalized stigma can be mitigated by a self-help group [54]. The members do not feel alone, which reduces their internalized stigma [55]. Members may also learn from their peers how to overcome internalized stigma [51]. Members may encourage each other during their group meetings to fight against stigma [16].

A self-help group is a prominent approach for mental health and substance use problems [14]. While a self-help group and its peer-to-peer counseling are not an inferior alternative to professional mental health support, it is beneficial for the participants to receive professional training from experts in mental health. They can learn how to deal with depressive symptoms and internalized stigma as well as expand social support. In Myanmar, mental health professionals are available in the central and some limited community levels, and their number is scarce [56,57]. Therefore, it would be feasible that people living with HIV in self-help groups and staff in the ART clinics receive training, rather than they find professionals who can be dispatched to the clinics.

## Limitations

Several limitations should be considered to interpret the findings of this study. First, this study was conducted under a cross-sectional design, and it cannot examine exposure and other variables as the causes of depressive symptoms. Still, this study provided baseline data for people living with HIV in Yangon city, Myanmar, regarding the participation in self-help groups and its associated factors. Second, ART clinics and people living with HIV were purposively selected by NAP and staff at the ART clinics without involvement by the authors. The authors could not keep track of the number of people living with HIV recruited and refused to participate in this study. Also, the number of participants per clinic did not reflect the sizes of the clinics. The majority recruited in this study were from NGO type clinics. A concern may arise if the study participants represented people living with HIV at the study site.

Third, depressive symptoms were self-reported; the reporting of depressive symptoms might be subject to social norms and pressures that people living with HIV perceived. To reduce this bias, interviews were conducted to ensure the anonymity of responses to interview questions. This study measured the different types of social support using the validated scale. However, social support could not be interpreted as a cause of depressive symptoms under the study design of this study. The investigation of reasons behind a lower level of social support among self-help group members was beyond our scope. Future studies should be designed to examine the role of social support in the causal relationship between the participation in self-help groups and depressive symptoms.

Lastly, the Cronbach's alpha of the AIDS Related Stigma Scale (0.68) was low, which might question this study's reliability. Limited number of internalized stigma scale was validated in Myanmar context.

## Conclusion

This study did not find evidence on the association between membership in self-help groups and depressive symptoms among people living with HIV who received care at ART clinics in Yangon city, Myanmar. In this study, although the mean level of social support was lower among self-help group members than non-members, social support was a protective factor against depressive symptoms regardless of the membership in self-help groups. While this study did not determine causality, the activities of self-help groups and care provided by the ART clinics might not be sufficient to address depressive symptoms among people living with HIV in the study site. Given the limited availability of mental health support services, self-help groups should be strengthened as an approach to address mental health and substance use problems among people living with HIV. Self-help groups and their members may need additional professional support to enhance their roles against depressive symptoms and stress.

As the level of social support was low, particularly among the members of self-help groups, people living with HIV should be encouraged to gain more social support through the

activities of self-help groups and the services at the ART clinics. These findings should be shared among NAP and NGO clinics to accelerate their activities to mitigate depressive symptoms among people living with HIV.

## Supporting information

**S1 Appendix. Information and data of people living with HIV.**
(XLSX)

## Acknowledgments

We would like to thank all people living with HIV for their voluntary participation in this study, and the Myanmar Positive Group for supporting this study.

## Author Contributions

**Conceptualization:** Myat Wint Than, Nicholus Tint Zaw, Masamine Jimba, Akira Shibanuma.

**Data curation:** Myat Wint Than, Nicholus Tint Zaw.

**Formal analysis:** Myat Wint Than, Akira Shibanuma.

**Funding acquisition:** Myat Wint Than, Masamine Jimba.

**Investigation:** Myat Wint Than, Nicholus Tint Zaw.

**Methodology:** Myat Wint Than, Junko Kiriya, Masamine Jimba, Hla Hla Win, Akira Shibanuma.

**Project administration:** Myat Wint Than, Nicholus Tint Zaw, Akira Shibanuma.

**Software:** Myat Wint Than, Akira Shibanuma.

**Supervision:** Kyi Minn, Yu Mon Saw, Junko Kiriya, Masamine Jimba, Hla Hla Win, Akira Shibanuma.

**Validation:** Myat Wint Than, Nicholus Tint Zaw, Kyi Minn, Yu Mon Saw, Junko Kiriya, Masamine Jimba, Hla Hla Win.

**Visualization:** Myat Wint Than.

**Writing – original draft:** Myat Wint Than.

**Writing – review & editing:** Nicholus Tint Zaw, Kyi Minn, Yu Mon Saw, Junko Kiriya, Masamine Jimba, Hla Hla Win, Akira Shibanuma.

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
