## [Decision Letter · Decision Letter 0]

8 Oct 2020

PONE-D-20-24440

Association between being a member of self-help group and depressive symptoms among people living with HIV in Yangon, Myanmar

PLOS ONE

Dear Dr. Shibanuma,

Thank you for submitting your manuscript to PLOS ONE. After careful consideration, we feel that it has merit but does not fully meet PLOS ONE’s publication criteria as it currently stands. Therefore, we invite you to submit a revised version of the manuscript that addresses the points raised during the review process.

We look forward to receiving your revised manuscript.

Kind regards,

Siyan Yi, MD, MHSc, PhD

Academic Editor

PLOS ONE

Additional Editor Comments:

We have received comments from four reviewers with contradicting opinions. One reviewer recommended 'rejection,' two suggested minor revisions, and one suggested major revisions. Please address each comment carefully. Please also take this opportunity to improve the paper as much as possible, including the quality of writing. Please ensure that the paper is aligned with the journal's guidelines and free from grammatical errors and typos. We will decide on whether to consider the manuscript further upon receiving the revised manuscript.

Journal Requirements:

Reviewers' comments:

Reviewer's Responses to Questions

**Comments to the Author**

1. Is the manuscript technically sound, and do the data support the conclusions?

Reviewer #1: Yes

Reviewer #2: Partly

Reviewer #3: Partly

Reviewer #4: Partly

2. Has the statistical analysis been performed appropriately and rigorously? 

Reviewer #1: Yes

Reviewer #2: No

Reviewer #3: No

Reviewer #4: Yes

3. Have the authors made all data underlying the findings in their manuscript fully available?

Reviewer #1: Yes

Reviewer #2: Yes

Reviewer #3: Yes

Reviewer #4: Yes

4. Is the manuscript presented in an intelligible fashion and written in standard English?

Reviewer #1: Yes

Reviewer #2: Yes

Reviewer #3: Yes

Reviewer #4: Yes

5. Review Comments to the Author

Reviewer #1: Dear Author,

Thank you for submitting this article. Here are my comments that you may want to consider:

Introduction

1. In Page 4, you defined adults as aged 15 and above. Please justify if this definition is according to national law in Myanmar.

2. Study design: to consider using the term of “comparative cross-sectional design”

3. Please explain whether the 19-item Medical Outcome Study Social Support Survey (MOS-SSS) is available in Burmese version.

4. Explanation of the clinics or hospital where the study was conducted is needed – this can be done by adding a section on study setting.

5. A clear explanation on participants recruitment (how those participants were recruited) is required.

6. In page 9, you mentioned: “of the members, 89 PLHIV (44.3%) and 86 PLHIV (32.7%) of the non-members had depressive symptoms.” This sentence is unclear and need to be re-write.

7. In page 13, title of Table 3 should be corrected.

8. In page 15, you mentioned: “Self-help group members were more likely to have depressive symptoms compared with the non-members”. Please explain implications of this findings in the discussion.

9. In page 16, you mentioned: “Among both members and non-members, women were more likely than men to have depression”. Further explanation on this finding is required.

10. There should be a discussion on implications of the study findings.

11. In page 18, you mentioned “PLHIV should check their mental health status during their visit to ART clinics”. Please explain how this can be possible. Please also include a description on the availability of the assessment for PLHIV in your study setting.

Reviewer #2: The self-help group and non self-help group are not comparable, so it is looks like the author could not compare the risk factors of depression in the two groups. This is a big problem in study design.

Social support is a part of self help. Why is social support lower in self help group? Does it mean that self-help has no effect?

In table 3, N=xxx ?

Reviewer #3: This study investigates the association of being a member of a self-help group and depressive symptoms among people living with HIV in Myanmar. Although the contribution of this paper is substantial, there is room for improvement by addressing the following comments:

Major comments:

Methods

1. Provide context-specific information regarding the organization of self-help group in the selected ART clinics. Is there any difference in the way the self-help group is organized in the selected ART clinics that could affect the results of this study?

2. Please include how many patients were receiving services from each of the selected ART clinics.

3. Provide the criteria that guided the purposive selection of the three selected ART clinics

4. Was refusal to participation or non-response documented for this study?

Analysis

5. The multiple logistic regression should include the variable “type of clinics” to control for its potential effect. Given that the majority of patients (82.5%) were recruited from a single ART clinic (NGO), I suggest that this variable be dichotomized into: NGO and Other clinics (NAP; CBO; Other).

Results

6. More than of half the participants were female. Does this proportion reflect the epidemic of HIV in Yangong?

7. The majority of patients (82.5%) were recruited from a single ART clinic (NGO). It is likely that the current results reflect the practices of this single ART clinic. The authors should elaborate more on this in the discussion section.

8. There are a number of differences (education; LGBT; internalized stigma) in the stratified analyses by member status. It is important to control for the variable “type of clinics” to explore its potential role in those documented differences in factors associated with depression my self-help group membership status.

Discussion

9. The second sentence of the discussion “Particularly, those who registered in a government-registered or community-based organization clinic were at higher risk” is not supported by the reported results.

10. Second paragraph: The second explanation regarding the self-help group members were more likely to depressive symptoms appears to be far-fetched. Particularly, the sentence “If the members improve their depressive symptoms they could share their experience with the non-members”. Assuming that experience-sharing affect the non-members; it should also affect the members who have joined the self-help group.

Conclusion

11. “Participation in self-help group was not enough to mitigate depressive symptoms although it was found to mitigate internalized stigma”. This sentence implies the notion of causality. This is cross-sectional study; the documented associations cannot infer causality. I suggest the authors to rephrase the sentence and keep to the word “association” and avoid the verb “to mitigate”.

Minor comments

12. Any explanation why the being a LGBT was associated with depressive symptoms only among non-members?

13. Why did you exclude patients with TB co-infection?

14. The cross-sectional design should be cited among the limitations of this study. With this design, it is not possible to ascertain causality of the documented associations.

15. Please include line numbers to facilitate review of the manuscript.

Reviewer #4: Comments for PONE-D-20-24440

Association between being a member of self-help group and depressive symptoms among people living with HIV in Yangon, Myanmar

The study explores the association between self-help group membership and depressive symptoms among PLHIV in Yangon, Myanmar. The study concluded that being a member/having a membership is positively correlated with having depressive symptoms.

Major limitation of the study:

Being a member, does not translate into or warrant that the person is actually having an active participation. The study did not measure the frequency of attending the self-help group meetings, consistency of participation, or the actual role/activity that the participants are involved in, which may provide a clearer picture of the “sense of belonging” to the self-help group, for us to be able to draw an association between participation and depressive symptoms. There is no measure of “sense of belonging” or level of involvement to the group. Hence, the conclusion (as stated in the of the abstract) that “participation in a self-help group was not enough to mitigate depressive symptoms” might be invalid, as the study did not measure the actual participation but a membership status only. As the authors have stated on page 16 line 4, “The quality of the membership experience may be more important”, yet there was no attempt by the authors to measure the quality of the membership experience rather than measuring the membership status only. Furthermore, the conclusion that highlighted the importance of “additional perceived social support” was too general and does not provide meaningful guideline for public health intervention. It would have been interesting to rather see “where” can we actually intervene; to strengthen the support from medical personnels? or family? or friend? or the self-help group?

Minor comments:

Methods:

1.“A two-stage sampling”, may imply stratified and clustered sampling which requires rigorous complex sample analyses. If the authors do not intend to carry out complex sampling analyses, I would suggest that the term is omitted from the manuscript.

2. Has the MOS-SSS been validated in Burmese?

3. Was there an incentive to participate in the study? What were the benefits of participating in the study?

4. Participants who are single and separated/widow may have different characteristics and hence, a different prevalence of having depressive symptoms. Not sure why they are grouped together.

5. Age grouped in age-group category rather than age as a continuous variable. Similarly, for an easier interpretation, would it be possible for the authors to categorised other continuous variable such as perceived social support (Low/High), Internalized stigma (Low/High), etc?

6. Was there an association between perceived social support and internationalized stigma? Would the authors suggest there is an interaction between having depressive symptoms, perceived social support, and internalized stigma?

Discussion:

Second paragraph does not at all provide a good argument for the counterintuitive finding; “First, being depressed might cause PLHIV to join self-help groups..” and “Second, the members who had improved….invite depressed peers to join their self-help group”, however, the participants in this study have been in the self-help group for over a year (that was the inclusion criteria), they are not new members to the group. Similarly, page 16 last sentence, these participants have been in the group for over a year.

Language:

Generally well-written, though improvements can be made. Grammatical errors here and there. Will leave this to the editorial office.

6. PLOS authors have the option to publish the peer review history of their article (what does this mean?). If published, this will include your full peer review and any attached files.

Reviewer #1: No

Reviewer #2: No

Reviewer #3: No

Reviewer #4: **Yes: **TECHASRIVICHIEN Teeranee

---

## [Author Response · Author response to Decision Letter 0]

20 Nov 2020

(We attach "Responses to reviewers" file, which has the same contents using a table format.)

Thank you very much for taking the time to review the manuscript. We responded to your valuable comments as below:

Reviewer #1

1

(Comment) In Page 4, you defined adults as aged 15 and above. Please justify if this definition is according to national law in Myanmar.

(Response) In Myanmar, different definitions of an adult exist. In Myanmar Penal Code, aged 15 and above is regarded as an adult and with the parent’s consent, a girl with the age of 15 can marry. However, Myanmar has already ratified Child Right Convention, and Ministry of Social Welfare defines the age of 18 and above as an adult. To avoid possible confusion, we rewrite the sentence without using the word “adults.”

We amended the sentence as follows:

“In 2018, HIV prevalence was estimated to be 0.8% among those who aged between 15 and 49. In total, 170,000 people received ART out of 240,000 PLHIV in Myanmar.” (line 29-31, page 4)

2

(Comment) Study design: to consider using the term of “comparative cross-sectional design”

(Response) We added the term “comparative” to the design. 

“A comparative cross-sectional study was conducted in Yangon, Myanmar in 2017.” (line 59, page 5)

3

(Comment) Please explain whether the 19-item Medical Outcome Study Social Support Survey (MOS-SSS) is available in Burmese version.

(Response) The Burmese version of Medical Outcome Study Social Support Survey (MOS-SSS) was used in this study. This MOS-SSS Burmese version has already been applied to previous studies in refugee camps along Thai-Burma Border (e.g., Reference number 45). Burmese is the official language in Myanmar and the majority use this language. We inserted the following paragraph in the manuscript. 

“The Cronbach’s alpha coefficients for subscales in the Burmese version were 0.87 on the emotional subscale, 0.85 on the tangible subscale, 0.87 on the affectionate subscale, and 0.82 on the positive social interaction subscale.” (line 125-127, page 8)

4

(Comment) Explanation of the clinics or hospital where the study was conducted is needed – this can be done by adding a section on study setting.

(Response) We explained the ART clinic in this study as follows:

“Study setting

This study was conducted in the venues of ART clinics. A structured questionnaire was administered to all of the participants through face-to-face interviews in the Burmese language by trained interviewers in private rooms at the ART clinics. The ART clinics included in this study offered HIV testing and prevention awareness sessions. When people were diagnosed with HIV positive, the clinic provided ART and counseling. ART clinics provided meeting space for self-help group members to discuss their challenges. Self-help groups were independent of ART clinics that provided treatment for them.” (line 66-74, page 5-6)

5

(Comment) A clear explanation on participants recruitment (how those participants were recruited) is required.

(Response) We added an extra explanation of how PLHIV were recruited at ART clinic. 

“Second, PLHIV were recruited by a convenience sampling method when they visited the clinics during the data collection period. The clinic reception staff announced this study in the waiting area. Those who were interested in the participation received explanations from the staff on the purpose of this study. Only those who were willing to participate in this study were sent to the interview rooms.” (line 89-95, page 6-7)

6

(Comment) In page 9, you mentioned: “of the members, 89 PLHIV (44.3%) and 86 PLHIV (32.7%) of the non-members had depressive symptoms.” This

sentence is unclear and need to be re-write.

(Response) We amended the sentence to improve the clarity as follows: 

“Depressive symptoms were found among 89 members (44.3%) and 86 non-members (32.7%).” (line 167-168, page 10)

7

(Comment) In page 13, title of Table 3 should be corrected.

(Response) We corrected the title of Table 3 as per your comment.

Table 3: Factors associated with having depressive symptoms stratified by the membership status of a self-help group (N=464) (line 212-213, page 14)

8

(Comment) In page 15, you mentioned: “Self-help group members were more likely to have depressive symptoms compared with the non-members”.

Please explain implications of this findings in the discussion.

(Response) As a result of adding the “type of clinic” variable in our regression model in Table 2 (by reflecting other reviewers’ comments), self-help group membership status is no longer statistically significant. We updated the discussion for implication from the finding. 

“Being self-help group members was not associated with having depressive symptoms. However, members who registered at NGO clinics had a lower risk of having depressive symptoms, compared to members who registered at NAP/CBO clinics. The NAP/CBO clinics had fewer human and material resources than NGO funded clinics. Some members who registered at NAP/CBO clinics volunteered in these clinics. Some members who registered at NAP/CBO clinics volunteered in these clinics, for example, for tracing the lost-to-follow-up of PLHIV. This might cause stress and burden on the members. Health professionals at NAP/CBO clinics may need an opportunity to learn from NGO clinics regarding how NGOs were supporting self-help groups.” (line 227-235, page 16) 

9

(Comment) In page 16, you mentioned: “Among both members and non-members, women were more likely than men to have depression”. Further

explanation on this finding is required.

(Response) Male-dominated social norm has prevailed in Myanmar at the household, community, and even state level. LGBT often face an issue of acceptance in the society. Consequently, people tend to blame women and LGBT, rather than men, among those infected with HIV. 

“Under persistent social norm related to gender, once women and LGBT are infected with HIV, they may perceive a higher level of stress than men. To address their stress, health professional needs to take proactive roles in raising gender equality awareness in the community and encouraging family members of PLHIV to respect gender diversity.” (line 249-253, page 17)

10

(Comment) There should be a discussion on implications of the study findings.

(Response) We modified the entire Discussion and Conclusion sections to add implications from the study findings. Point-to-point modifications were described as the responses to other comments in this file.

11

(Comment) In page 18, you mentioned “PLHIV should check their mental health status during their visit to ART clinics”. Please explain how this can

be possible. Please also include a description on the availability of the assessment for PLHIV in your study setting.

(Response) We added a new paragraph for a possible solution to improve PLHIV’s mental health status under scarcity in the professional resources.

“A self-help group is a prominent approach for mental health and substance use problems. While a self-help group and its peer-to-peer counselling are not an inferior alternative to professional mental health support, it is beneficial for the participants to receive professional training from experts in mental health. They can learn how to deal with depressive symptoms and internalized stigma as well as expand social support. In Myanmar, mental health professionals are available only in the central and some limited community levels, and their number is scarce. Therefore, it would be feasible that PLHIV in self-help groups and staff in the ART clinics receive training, rather than they find professionals who can be dispatched to the clinics.” (line 276-284, page 18)

Reviewer #2

1

(Comment) The self-help group and non self-help group are not comparable, so it is looks like the author could not compare the risk factors of depression in the two groups. This is a big problem in study design.

(Response) We recruited PLHIV from three ART clinics regardless of their participation in a self-help group. Then, we asked if PLHIV joined a self-help group or not by a questionnaire survey. In this sense, we believe that we can investigate if participation in a self-help group was associated with depressive symptoms. 

Moreover, we examined factors associated depressive symptoms using a pooled analysis that included members and non-members in the same dataset, in addition to the stratified models by the membership status. These models were used to confirm if membership status might affect the association between other covariates and the outcome. Since the background characteristics of members and non-members were different, these characteristics were controlled in multiple regression models. As this is the cross-sectional study, we did not address a causal relationship, and we mentioned it in the limitation section. 

We amended the title to the following: “Assessing depressive symptoms among people living with HIV in Yangon, Myanmar: Does being a member of self-help group matter?” 

2

(Comment) Social support is a part of self help. Why is social support lower in self help group? Does it mean that self-help has no effect?

(Response) We used Medical Outcome Study Social Support Survey (MOS-SSS) scale to measure the level of social support. Some items in MOS-SSS, particularly emotional-informational support, could be improved through the participation in a self-help group. However, in this study, the level of social support was lower among self-help group members. PLHIV who have higher social support might not join self-help group as they might receive social support from family members or friends or colleagues. Those who needed social support might join to seek social support. This study used the cross-sectional design and it did not establish a causal relationship. We recommended a longitudinal study to examine the effect of self-help groups and social support among self-help group members for future. 

We explained the source of social support in the limitation section as follows. 

“This study measured the different types of social support using the validated scale. However, social support could not be interpreted as a cause of depressive symptom under the study design of this study. Investigate reasons behind the lower level of social support among self-help group members was beyond our scope. Future studies should be designed to examine the role of social support in the causal relationship between the participation in self-help groups and depressive symptoms.” (line 296-301, page 18-19)

3

(Comment) In table 3, N=xxx ?

(Response) We corrected the title of Table 3.

“Table 3: Factors associated with having depressive symptoms stratified by the membership status of a self-help group (N=464)” (line 212-213, page 14)

Reviewer #3

1

(Comment) 

Methods

Provide context-specific information regarding the organization of self-help group in the selected ART clinics. Is there any difference in the way the self-help group is organized in the selected ART clinics that could affect the results of this study?

(Response) We added context-specific information regarding the organization of self-help groups in the newly added section of “Study setting.”

“This study was conducted in the venues of ART clinics. A structured questionnaire was administered to all of the participants through face-to-face interviews in Burmese language by trained interviewers in private rooms at the ART clinics. The ART clinics included in this study offered HIV testing and prevention awareness sessions. When people were diagnosed with HIV positive, the clinic provided ART and counseling. ART clinics provided meeting space for self-help group members to discuss their challenges. Self-help groups were independent of ART clinics that provided treatment for them.” (line 66-74, page 5-6)

2

(Comment) Please include how many patients were receiving services from each of the selected ART clinics.

(Response) To protect confidentiality of their clients, ART clinics did not provide detailed information about patients. We could not include information about the number of patients at the selected ART clinics.

3

(Comment) Provide the criteria that guided the purposive selection of the three selected ART clinics

(Response) We added explanations according to your comment. Assistant Director from National AIDS Programme (NAP) at the central level recommended three ART clinics that cooperated well with NAP and sent a regular report to NAP at both state and central levels.

“First, three ART clinics were purposively selected through consultation with the National AIDS Programme (NAP). Assistant Director from NAP recommended three ART clinics that cooperated well with NAP and sent a regular report to NAP.” (line 86-89, page 6)

This purposive selection was indicated as a limitation as follows:

“Second, PLHIV were purposively selected by staff at the ART clinics, without involvement by the authors. That is, data on the number of recruited and refused to participate in the study were not available. Also, the number of PLHIV participated at the ART clinics did not reflect the sizes of the clinics. The majority of PLHIV recruited in this study was from NGO type clinics. These may affect the representativeness of this study.” (line 289-294, page 18)

4

(Comment) Was refusal to participation or non-response documented for this study?

(Response) Due to the procedure of participant recruitment, we could not obtain the information about any refusal. The interviewers waited at the interview room, and the receptionist from the ART clinics sent those who agreed to participate in the interview. We modified explanations about recruitment as follows: 

“Second, PLHIV were recruited by a convenience sampling method when they visited the clinics during the data collection period. The clinic reception staff announced this study in the waiting area. Those who were interested in the participation received explanations from the staff on the purpose of this study. Only those who were willing to participate in this study were sent to the interview rooms.” (line 89-95, page 6-7)

We also explain this as a limitation. 

“Second, PLHIV were purposively selected by staff at the ART clinics, without involvement by the authors. That is, data were not available on the number of recruited and refused to participate in the study. Also, the number of PLHIV participated at the ART clinics did not reflect the sizes of the clinics. The majority of PLHIV recruited in this study was from NGO type clinics. These may affect the representativeness of this study.” (line 289-294, page 18)

5

(Comment) Analysis

(Response) The multiple logistic regression should include the variable “type of clinics” to control for its potential effect. Given that the majority of patients (82.5%) were recruited from a single ART clinic (NGO), I suggest that this variable be dichotomized into: NGO and Other clinics (NAP; CBO; Other).

We added the variable “type of clinic” in our analysis. We dichotomized this variable as suggested. 

“In addition, the variable “type of clinic” was recorded as the organization of the ART clinic where PLHIV was recruited. It was used as dichotomized: NGO clinic and other (clinic run by NAP or a community-based organization).” (line 129-131, page 8)

6

(Comment) 

Results

More than of half the participants were female. Does this proportion reflect the epidemic of HIV in Yangong?

(Response) Among 54,515 PLHV, 24,132 (44.3%) were women with PLHIV in Yangon. This was reported in “Progress Report 2018” by the National AIDS Programme, Ministry of Health and Sports, Myanmar. This discrepancy was inevitable in the study design and explained as a limitation. 

“Second, PLHIV were purposively selected by staff at the ART clinics, without involvement by the authors. That is, data on the number of recruited and refused to participate in the study were not available. Also, the number of PLHIV participated at the ART clinics did not reflect the size of the clinics. The majority of PLHIV recruited in this study was from NGO type clinics. These may affect the representativeness of this study.” (line 289-294, page 18)

7

(Comment) The majority of patients (82.5%) were recruited from a single ART clinic (NGO). It is likely that the current results reflect the practices of this single ART clinic. The authors should elaborate more on this in the discussion section.

(Response) We explained the practices of clinics in the Discussion session. 

“Being self-help group members was not associated with having depressive symptoms. However, members who registered at NGO clinics had a lower risk of having depressive symptoms, compared to members who registered at NAP/CBO clinics. The NAP/CBO clinics had fewer human and material resources than NGO funded clinics. Some members who registered at NAP/CBO clinics volunteered in these clinics, for example, for tracing the lost-to-follow-up of PLHIV. This might cause stress and burden on the members. Health professional at NAP/CBO clinics may need an opportunity to learn from NGO clinics regarding how NGOs were supporting self-help groups.” (line 227-235, page 16) 

The majority of PLHIV (82,5%) were recruited from NGO running clinics, and this limited the representativeness of this study. We explained this limitation in limitation section as follows:

“Second, PLHIV were purposively selected by staff at the ART clinics, without involvement by the authors. That is, data on the number of recruited and refused to participate in the study were not available. Also, the number of PLHIV participated at the ART clinics did not reflect the size of the clinics. The majority of PLHIV recruited in this study was from NGO type clinics. These may affect the representativeness of this study.” (line 289-294, page 18)

8

(Comment) There are a number of differences (education; LGBT; internalized stigma) in the stratified analyses by member status. It is important to control for the variable “type of clinics” to explore its potential role in those documented differences in factors associated with depression my self-help group membership status.

(Response) We added the variable “type of clinic” in our analysis. Please see updated result and Table 3.

9

(Comment) The second sentence of the discussion “Particularly, those who registered in a government-registered or community-based organization clinic were at higher risk” is not supported by the reported results.

(Response) We accepted this comment and removed this sentence from the manuscript.

10

(Comment) Second paragraph: The second explanation regarding the self-help group members were more likely to depressive symptoms appears to be far-fetched. Particularly, the sentence “If the members improve their depressive symptoms they could share their experience with the non-members”. Assuming that experience-sharing affect the non-members; it should also affect the members who have joined the self-help group.

(Response) Reflecting the change in the model that included the variable “type of clinic,” being members of self-help groups were no longer statistically significant. We removed the part of discussion about this significance.

11

(Comment) Conclusion

(Response) “Participation in self-help group was not enough to mitigate depressive symptoms although it was found to mitigate internalized stigma”. This sentence implies the notion of causality. This is cross-sectional study; the documented associations cannot infer causality. I suggest the authors to rephrase the sentence and keep to the word “association” and avoid the verb “to mitigate”.

We removed a sentence that implied the causality from the Conclusion section and modified descriptions in the entire Conclusion section.

12

(Comment) Minor comments

Any explanation why the being a LGBT was associated with depressive symptoms only among non-members?

(Response) We modified the discussion regarding gender and depressive symptoms, including the description of LGBT as follows:

“HIV-infected women and LGBT face stigma and discrimination in their daily lives. Under persistent social norm related to gender, once women and LGBT are infected with HIV, they may perceive a higher level of stress than men. To address their stress, health professional needs to take proactive roles in raising gender equality awareness in the community and encouraging family members of PLHIV to respect gender diversity.” (line 248-253, page 17)

13

(Comment) Why did you exclude patients with TB co-infection?

(Response) Most (70%) of the interviewers were also PLHIV in this study. PLHIV with TB co-infection was excluded from this study to prevent transmission. 

14

(Comment) The cross-sectional design should be cited among the limitations of this study. With this design, it is not possible to ascertain causality of the documented associations.

(Response) We added more explanation in the limitation section regarding the cross-sectional design of this study. 

“First, this study was conducted under a cross-sectional design and it cannot examine exposure and other variables as the causes of depressive symptoms.” (line 286-288, page-18)

15

(Comment) Please include line numbers to facilitate review of the manuscript.

(Response) We include the line numbers in the amended manuscript. 

Reviewer #4

1

(Comment) Major limitation of the study:

Being a member, does not translate into or warrant that the person is actually having an active participation. The study did not measure the frequency of attending the self-help group meetings, consistency of participation, or the actual role/activity that the participants are

involved in, which may provide a clearer picture of the “sense of belonging” to the self-help group, for us to be able to draw an association between participation and depressive symptoms. There is no measure of “sense of belonging” or level of involvement to the group.

Hence, the conclusion (as stated in the of the abstract) that “participation in a self-help group was not enough to mitigate depressive symptoms” might be invalid, as the study did not measure the actual participation but a membership status only. As the authors have stated on page 16 line 4, “The quality of the membership experience may be more important”, yet there was no attempt by the authors to measure the quality of the membership experience rather than measuring the membership status only. Furthermore, the conclusion that highlighted the importance of “additional perceived social support” was too general and does not provide meaningful guideline for public health intervention. It would have been interesting to rather see “where” can we actually intervene; to strengthen the support from medical personnels? or family? or friend? or the self-help group?

(Response) This study did not use the collected data on the frequency of attending self-help group meetings and activities. However, the activities of self-help groups were not identical. Some groups have been formed for one decade while others have been for one year. Depending on the relationships with donor agencies, some groups have funding while others have not received anything. Therefore, based on the observations of self-help groups, we judged that the frequency of attending self-help group meetings and the length of membership might not be a proxy for the exposure to the activities or the sense of belongings.

Regarding our implications on enhancing social support, we modified the Conclusion section as follows:

“In this study, self-help groups served as venues where PLHIV interacted with each other and discussed their challenges, including those who had depressive symptoms and a low level of social support.” (line 308-311, page 19)

“As the level of social support was low, particularly among the members of self-help groups, PLHIV should be encouraged to gain more social support through the activities of self-help groups and the services at the ART clinics. These findings should be shared among NAP and NGO clinics to accelerate their activities to mitigate depressive symptoms among PLHIV. Further prospective longitudinal studies are needed to follow up changes in depressive symptoms after participation in a self-help group.” (line 323-329, page 19-20)

2

(Comment) Minor comments:

Methods:

1.“A two-stage sampling”, may imply stratified and clustered sampling which requires rigorous complex sample analyses. If the authors do not

intend to carry out complex sampling analyses, I would suggest that the term is omitted from the manuscript.

(Response) We have removed this “A two-stage sampling” from the manuscript. 

3

(Comment) Has the MOS-SSS been validated in Burmese?

(Response) The Burmese version of Medical Outcome Study Social Support Survey (MOS-SSS) was used in our study. This MOS-SSS has already been applied in refugee camps along Thai-Burma Border. Not only Burmese but also Karen versions are available. We inserted the following paragraph in the manuscript. 

“The Cronbach’s alpha coefficients for subscales in the Burmese version were 0.87 on the emotional subscale, 0.85 on the tangible subscale, 0.87 on the affectionate subscale, and 0.82 on the positive social interaction subscale.” (line 125-127, page 8)

4

(Comment)Was there an incentive to participate in the study? What were the benefits of participating in the study?

(Response) To compensate time for the interview, small incentive (soap and towel) was presented the participants. 

“PLHIV who participated in this study received soap and towel at the end of the interview.” (line 94-95, page 7)

5

(Comment) Participants who are single and separated/widow may have different characteristics and hence, a different prevalence of having depressive

symptoms. Not sure why they are grouped together.

(Response) We understand that non-marital statuses have different forms, namely, single, separated, and widowed. In a large-scale survey, these could have different categories. However, since the sample size is not large in this study, the number of PLHIV that were fallen into each of the non-marital status categories were small. Consequently, independent categories for single, separated, and widowed might capture the unobserved characteristics of certain individuals had, rather than the non-marital status of single, separated, or widowed themselves. And in a multiple regression analysis, having a category with a few individuals might result in very wide confidence intervals or inability of estimation. Therefore, we decided to integrate these non-marital categories into a single integrated category. 

Single, separated, and widowed have common characteristics in a sense that they could not seek for support from their partners. In Myanmar, single, separated, and widowed person tend to live together with sibling or parents and could seek support from these family members. 

6

(Comment) Age grouped in age-group category rather than age as a continuous variable. Similarly, for an easier interpretation, would it be possible for the authors to categorised other continuous variable such as perceived social support (Low/High), Internalized stigma (Low/High), etc?

(Response) We understand the advantage of converting a continuous variable into a dichotomized variable and that health scientists often prefer dichotomization. However, prominent statisticians do not necessarily support the dichotomization due to the loss of rich information that the original continuous variable has. 

Please refer to the following article:

Altman DG et al. The cost of dichotomising continuous variables. BMJ. 2006 May 6; 332(7549):1080.

7

(Comment) Was there an association between perceived social support and internationalized stigma? Would the authors suggest there is an interaction between having depressive symptoms, perceived social support, and internalized stigma?

(Response) Among covariates in the regression model in this study, perceived social support was negatively associated with internalized stigma in bivariate analysis. However, since the interaction term between social support and internalized stigma were not statistically significant, we decided not to incorporate an interaction term between them. 

8

(Comment) Discussion:

Second paragraph does not at all provide a good argument for the counterintuitive finding; “First, being depressed might cause PLHIV to join self-help groups..” and “Second, the members who had improved….invite depressed peers to join their self-help group”, however, the participants in this study have been in the self-help group for over a year (that was the inclusion criteria), they are not new members to the group. Similarly, page 16 last sentence, these participants have been in the group for over a year.

(Response) After including “Type of clinic” variable in the regression analysis according to other reviewers’ comments, being self-help group members was not associated with having depressive symptoms. We rewrote the entire paragraph.

Under the cross-sectional design of this study, we do not have information on the pre-membership levels of depressive symptoms, social support, and internalized stigma. Nevertheless, based on the findings regarding the current levels of these measurements, we tried to list up possible reasons regarding what could happen before their membership, by referring to previous studies. Please kindly note that the list of possible reasons did not guarantee that it really happened in the study site.

9

(Comment) Language:

Generally well-written, though improvements can be made. Grammatical errors here and there. Will leave this to the editorial office.

(Response) We have checked grammatical errors and corrected them accordingly.

---

## [Decision Letter · Decision Letter 1]

19 Jan 2021

PONE-D-20-24440R1

Assessing depressive symptoms among people living with HIV in Yangon, Myanmar: Does being a member of self-help group matter?

PLOS ONE

Dear Dr. Shibanuma,

Thank you for submitting your manuscript to PLOS ONE. After careful consideration, we feel that it has merit but does not fully meet PLOS ONE’s publication criteria as it currently stands. Therefore, we invite you to submit a revised version of the manuscript that addresses the points raised during the review process.

We look forward to receiving your revised manuscript.

Kind regards,

Siyan Yi, MD, MHSc, PhD

Academic Editor

PLOS ONE

Additional Editor Comments:

**Editor’s comments**

We thank the authors for addressing outstanding comments from the reviewers. The revised manuscript has been much improved and almost ready for publication. In general, it is clear and easy to read. However, I have spotted several grammatical errors, typos, misuse of punctuations, and complex sentences across the manuscript. Many statements also require clarification. I believe it is worth spending a little more time cleaning them up. Here are some suggestions, which may not be exhaustive, and the revised manuscript requires thorough proofreading.

**Abstract**

As recommended by UNAIDS, people (such as PLHIV) should never be referred to as an abbreviation since this is dehumanizing. Instead, the name or identity of the group should be written out in full. Please see: https://www.unaids.org/sites/default/files/media_asset/2015_terminology_guidelines_en.pdf‘Data’ is plural form of ‘datum.’ Please use a plural verb with it; e.g., data were collected…Methods: ‘…as well as other factors’ may be unnecessary. Instead, please keep the space to briefly describe sampling methods.Please double-check 95% CI in this sentence: The membership was not associated with having depressive symptoms (adjusted odds ratio [AOR] 1.59, 95% confidence interval [CI] 1.98 - 2.59), which appeared incorrect.Conclusions: To save space for other essential information, the conclusions can be more condensed. For instance, the first and second sentences are too broad and do not deserve space.Please remove the keywords. We need them only in the submission system. Also, ‘people living with HIV’ should not be capitalized.

**Introduction**

First sentence: Not sure ‘chronically manageable’ is what the authors intended to say. Consider saying, ‘HIV infection has become a chronic, manageable condition.’Line 6: Please remove ‘In practice.’Lines 13-14: Please remove “In a self-help group of PLHIV,” as it is redundant and unnecessary.Line 19: Add ‘,’ before ‘possibly.’Line 21: Please remove ‘In addition to self-help groups’ as it does not go along well in the following statement on social support. Self-help groups are a form of social support.Line 26: ‘sub-classified.’Lines 26-28: The sentence “As the effect of perceived social support may differ from actual social support, perceived social support is more important in mental health” is not clear. It’s difficult to understand how the assumption that ‘perceived social support’ is more important in mental health is explained by the different effects between actual and perceived social support.Lines 29-30: Suggest revising, “In 2018, the estimated HIV prevalence among adults aged 15-49 in Myanmar was 0.8%.Lines 30-31: Please consider revising the sentence: “Of the total 240000 people living with HIV in Myanmar, xx% received ART in xxxx.”Lines 32-33: Please add ‘,’ before ‘and.’ Also, this information is not clear, “Despite these improvements in care, 60.0% of PLHIV feel ashamed and 18.0% are denied access to sexual and reproductive health services due to their HIV status in Myanmar.” What do people living with HIV feel ashamed of? By whom are they denied access to SRH services?Line 37: Please double check the measurement of depression. Was it clinical depression or depressive symptoms? 30% prevalence of depression seems too high. Also, when mentioning any ‘prevalence,’ please specify the population – was it a national prevalence?Line 40: Please remove or specify ‘To clarify that’ – what is ‘that?’

**Methods**

Line 59: …in Yangon, Myanmar, in 2017.Line 60: … from August to September 2017.Lines 60-61: Please revised this sentence: “Yangon was selected as the study site because this city reported many HIV cases.” Instead of saying “many HIV cases,” please provide the actual number of total cases or the proportion of people living with HIV in Yangon relative to the national people living with HIV population size.Please be consistent and specific: Is Yangon different from Yangon city?Line 65: What did the authors refer to by ‘public teams?’Line 67: “This study was conducted in ART clinics.” Please also indicate the number of ART clinics in the city and the study, and how they were selected.Lines 67-69: The sentence “A structured questionnaire was administered to all of the participants through face-to-face interviews in Burmese language by trained interviewers in private rooms at the ART clinics” should not be placed under the ‘Study setting.” Also, it was not clear whether ‘all participants’ refer to a ‘take-all” approach. If not, a sampling procedure should be described.Line 71: … ; the clinic provided ART and counseling. Also, please specify the ‘counseling’ – HIV confidential counseling and testing or other counseling services?Line 80: … and 12.8 for the standard care group. The calculation provided a required sample size of…Lines 81-82: It appeared that the inclusion criteria are not exhaustive. The wording for inclusion and exclusion criteria should be improved. Also, please provide the reason why people with HIV/tuberculosis co-infection were excluded.Line 86: In this study, the recruitment process is as follows…The first step in the participant recruitment process may introduce serious selection bias that affects the study population's representativeness. I would suggest highlighting it in the limitations.Lines 89-90: Please elaborate on how the ‘convenience sampling method’ was performed.Line 97: Please remove “To assess the membership status of self-help group [41],” which is unnecessary and somewhat redundant.Line 98: … being a member of a self-help group.Line 99: Please remove ‘presence of.’Line 103: Please replace ‘;’ by ‘.’Line 105: An answer to each question was given…Lines 107-108: The minimum mean score was one, and the maximum mean score was four.Line 116: How was ART adherence measured?Line 119: …, with a total score ranging from 0 to 7.Line 123: Please specify the value in the brackets (0.91). Is it a Cronbach’s alpha?Line 133: ‘Chi-square test’ should be capitalized.Line 137: Bivariate analyses were conducted… Also, please specify what the bivariate analyses are – are they different from the bivariate analyses mentioned above?Line 138: Only the variables associated with…Lines 138-140: The selection of the covariates for multiple regression analyses was not clear. What was the cut-off to define the significance level in bivariate analyses? Were all socio-demographic factors included in the models?Lines 147-149: Please provide a reference number for ethics approvals.Line 153-154: Confidentiality was maintained by not recording any personal identifiers in....

**Results**

Line 157: Please revise ‘PLHIV characteristics’ to ‘Participant characteristics.’For dichotomous variables, presenting only one group is sufficient.Line 162: …, and of non-members was…Line 169: …, which was lower…Line 170: Remove ‘,’ before ‘and 65.1’Table 1: Self-help group members, non-membersLine 194: The length of more than one year in a self-help group…Lines 194-195: Non-significant results may not be presented.Table 3: Self-help group members, non-members

**Discussion**

Line 222: However, self-help group members registered at NGO clinics…Line 223: … symptoms compared to members at…Line 228: However, members registered at…Line 229: … of having depressive symptoms compared to…Line 240: … and what duration they want…Line 245: In this study, gender differences existed in…Line 246: …non-members, consistent with previous studiesLine 249: Under persistent social norms related…Line 254: Please be consistent in the terminology use – depression or depressive symptomsLine 262: … symptoms, both among the members and non-membersLine 277: Please use consistent English – British or American; e.g., counselingThe writing quality of the limitations section needs improvement as it is the most difficult to read.Line 287: …, and it cannot…Line 292: Also, the number of PLHIV who participated atLine 299: …The investigation of reasons…The whole conclusions section requires improvement. The first few sentences are too broad and should summarize key findings. The third and fourth sentences are the same. Please avoid repeating results and discussions in this section.

**References**

The references need improvement as they are not consistent and not aligned with PLOS’ guidelines.

Reviewers' comments:

Reviewer's Responses to Questions

**Comments to the Author**

1. If the authors have adequately addressed your comments raised in a previous round of review and you feel that this manuscript is now acceptable for publication, you may indicate that here to bypass the “Comments to the Author” section, enter your conflict of interest statement in the “Confidential to Editor” section, and submit your "Accept" recommendation.

Reviewer #1: All comments have been addressed

Reviewer #3: All comments have been addressed

2. Is the manuscript technically sound, and do the data support the conclusions?

Reviewer #1: Yes

Reviewer #3: Yes

3. Has the statistical analysis been performed appropriately and rigorously? 

Reviewer #1: Yes

Reviewer #3: Yes

4. Have the authors made all data underlying the findings in their manuscript fully available?

Reviewer #1: Yes

Reviewer #3: Yes

5. Is the manuscript presented in an intelligible fashion and written in standard English?

Reviewer #1: Yes

Reviewer #3: No

6. Review Comments to the Author

Reviewer #1: Dear authors

I have read through your corrected manuscript and found that you have highlighted all my comments very well. Congratulations. Therefore, I have no issue to support publication of your paper in the journal.

Reviewer #3: The authors have convincingly addressed the comments. The manuscript has greatly improved, although it might still require a professional English language editing to improve the reporting style.

7. PLOS authors have the option to publish the peer review history of their article (what does this mean?). If published, this will include your full peer review and any attached files.

Reviewer #1: No

Reviewer #3: No

---

## [Author Response · Author response to Decision Letter 1]

5 Mar 2021

Thank you very much for giving us an opportunity of making revisions to the manuscript. Since we responded to a number of points, the entire responses to reviewers and editors were summarized in a separate file of "Response to Reviewers.rtf."

---

## [Editor Report · Decision Letter 2]

8 Mar 2021

Assessing depressive symptoms among people living with HIV in Yangon, Myanmar: Does being a member of self-help group matter?

PONE-D-20-24440R2

Dear Dr. Shibanuma,

We’re pleased to inform you that your manuscript has been judged scientifically suitable for publication and will be formally accepted for publication once it meets all outstanding technical requirements.

Kind regards,

Siyan Yi, MD, MHSc, PhD

Academic Editor

PLOS ONE
---

## [Editor Report · Acceptance letter]

10 Mar 2021

PONE-D-20-24440R2 

Assessing depressive symptoms among people living with HIV in Yangon city, Myanmar: Does being a member of self-help group matter? 

Dear Dr. Shibanuma:

I'm pleased to inform you that your manuscript has been deemed suitable for publication in PLOS ONE. Congratulations! Your manuscript is now with our production department. 

Kind regards, 

on behalf of

Dr. Siyan Yi 

Academic Editor

PLOS ONE